# Competitive interactions between culturable bacteria are highly non-additive

Amichai Baichman-Kass, Tingting Song, Jonathan Friedman*

Institute of Environmental Sciences, Hebrew University, Rehovot, Israel

**Abstract** Microorganisms are found in diverse communities whose structure and function are determined by interspecific interactions. Just as single species seldom exist in isolation, communities as a whole are also constantly challenged and affected by external species. Though much work has been done on characterizing how individual species affect each other through pairwise interactions, the joint effects of multiple species on a single (focal) species remain underexplored. As such, it is still unclear how single-species effects combine to a community-level effect on a species of interest. To explore this relationship, we assayed thousands of communities of two, three, and four bacterial species, measuring the effect of single, pairs of, and trios of 61 affecting species on six different focal species. We found that when multiple species each have a negative effect on a focal species, their joint effect is typically not given by the sum of the effects of individual affecting species. Rather, they are dominated by the strongest individual-species effect. Therefore, while joint effects of multiple species are often non-additive, they can still be derived from the effects of individual species, making it plausible to map complex interaction networks based on pairwise measurements. This finding is important for understanding the fate of species introduced into an occupied environment and is relevant for applications in medicine and agriculture, such as probiotics and biocontrol agents, as well as for ecological questions surrounding migrating and invasive species.

*For correspondence: yonatan.friedman@mail.huji.ac.il

Competing interest: The authors declare that no competing interests exist.

## Editor's evaluation

This important study presents an interesting example of how complexities of communities may be reduced by showing that when partner species exert a negative effect on the focal species, the joint effects are generally not additive, but rather dominated by the strongest single effect. The evidence, enabled by thousands of measurements using nanodroplet-based microfluidics, is compelling, although the generality of the conclusion awaits further studies. This paper is of interest to microbial ecologists and synthetic biologists.

## Introduction

Scarce are the environments on Earth not colonized by bacteria. In addition to naturally playing important roles from driving biogeochemical cycles at the ecosystem level (*Cavicchioli et al., 2019*; *Arrigo, 2005*; *Falkowski et al., 2008*) to supporting host health at the individual level (*Berendsen et al., 2012*; *Manor et al., 2020*; *Gilbert et al., 2018*), bacteria have also been harnessed for countless biotechnological applications across industries such as food preservation (*Motarjemi, 2002*), plant and animal health (*Berendsen et al., 2012*; *Júnior et al., 2021*), biocontrol of pathogens (*Köhl et al., 2019*), as well as decomposition of toxic compounds, and production of valuable ones (*Varjani et al., 2017*; *Ro et al., 2006*; *Fang and Smith, 2016*; *Mainka et al., 2021*). In natural environments, bacteria often form rich and complex communities, but understanding how these communities organize has

**eLife digest** Bacteria can be found almost everywhere on earth. Often, multiple species of bacteria live together in communities, which perform vital roles that affect everything from our health to the planet's ecosystems.

A single species within this community can sometimes be particularly important, for example if it is causing disease in its host or producing a vital nutrient. However, the other species within this community can influence the growth of this focus species, either by inhibiting or promoting it.

It is challenging to predict how a certain species is going to fare within a bacterial community as it remains partly unclear how groups of bacteria affect each other. Some theory suggests that adding up or averaging the influences of all the bacteria in a community would be the best way to predict what will happen.

To study these microorganism interactions, Baichman-Kass, Song and Friedman monitored thousands of bacterial communities, consisting of two to four different species. By using species that express fluorescent proteins, they were able to measure the abundance of the specific bacteria of interest in each of these communities.

Baichman-Kass et al. found that in communities where all the species were only competing with or supporting the bacteria of interest, the individual affecting species with the strongest effect dominated the combined effect. This 'strongest effect' model offered accurate predictions for the joint effects of competitive communities, however predicting outcomes in supporting communities proved more difficult. This could indicate that the mechanisms for supporting other species are more intricate than the means of competition.

The study of Baichman-Kass et al. brings us closer to understanding how the abundance of a given bacterium can be influenced through the actions of other bacterial species. Among other uses, it may be important in medicine, where it is desirable to reduce the amount of a bacteria that causes disease, or in agriculture where bacteria that protect plants from diseases and fungi, need to be boosted. Improving our ability to predict the outcome of introducing new species to an environment increases both the effectiveness and possible scope of such applications.

proven difficult (*Widder et al., 2016*). Elucidating the rules that govern microbial ecology can both offer insight into larger ecological systems and allow us to better manipulate and design microbial communities for the desired functions.

The structure of microbial communities is determined by the interactions between the involved species (*Konopka et al., 2015*; *Barbier et al., 2018*; *Qian and Akçay, 2020*). In recent years, much effort has been put into measuring pairwise interactions of different species from, and in, different environments (*Foster and Bell, 2012*; *Vetsigian et al., 2011*; *Kehe et al., 2021*). But it is still unclear to what extent the joint effects of multiple species on a focal species of interest (e.g., a pathogen) can be inferred from pairwise measurements. Such inference may be challenging due to the presence of indirect interactions: the affecting species may alter each other's abundances (termed interaction chains) or may modify each other's effect on the focal species (termed interaction modification, or higher-order interactions) (*Sanchez, 2019*; *Wootton, 2002*).

Despite a strong theoretical foundation, empirical studies in recent years have shown conflicting results regarding the importance of higher order interactions and indirect effects (*Levine et al., 2017*). For some functions, such as degradation of complex molecules (*Sanchez-Gorostiaga et al., 2019*; *Gralka et al., 2020*) and antibiotic production (*Tyc et al., 2014*; *Qi et al., 2021*; *Westhoff et al., 2021*), clear evidence of such interactions has been found. Furthermore, in both empirical and theoretical studies, the presence or absence of an additional species has been shown to affect interactions, and even the outcome of invasion and coexistence in some systems (*Mickalide and Kuehn, 2019*; *Chang et al., 2022*; *Hsu et al., 2019*). Additional theoretical work has shown that commonly used ecological models (i.e., generalized Lotka-Voltera) do not properly capture microbial community interactions, partially due to the nature of these interactions (chemically mediated as opposed to predator-prey) (*Momeni et al., 2017*). However, other studies have shown that both structures of, and interactions within, larger communities can be accurately predicted from pairwise interactions alone using variations of said models (*Friedman et al., 2017*; *Meroz et al., 2021*; *Guo and Boedicker*,

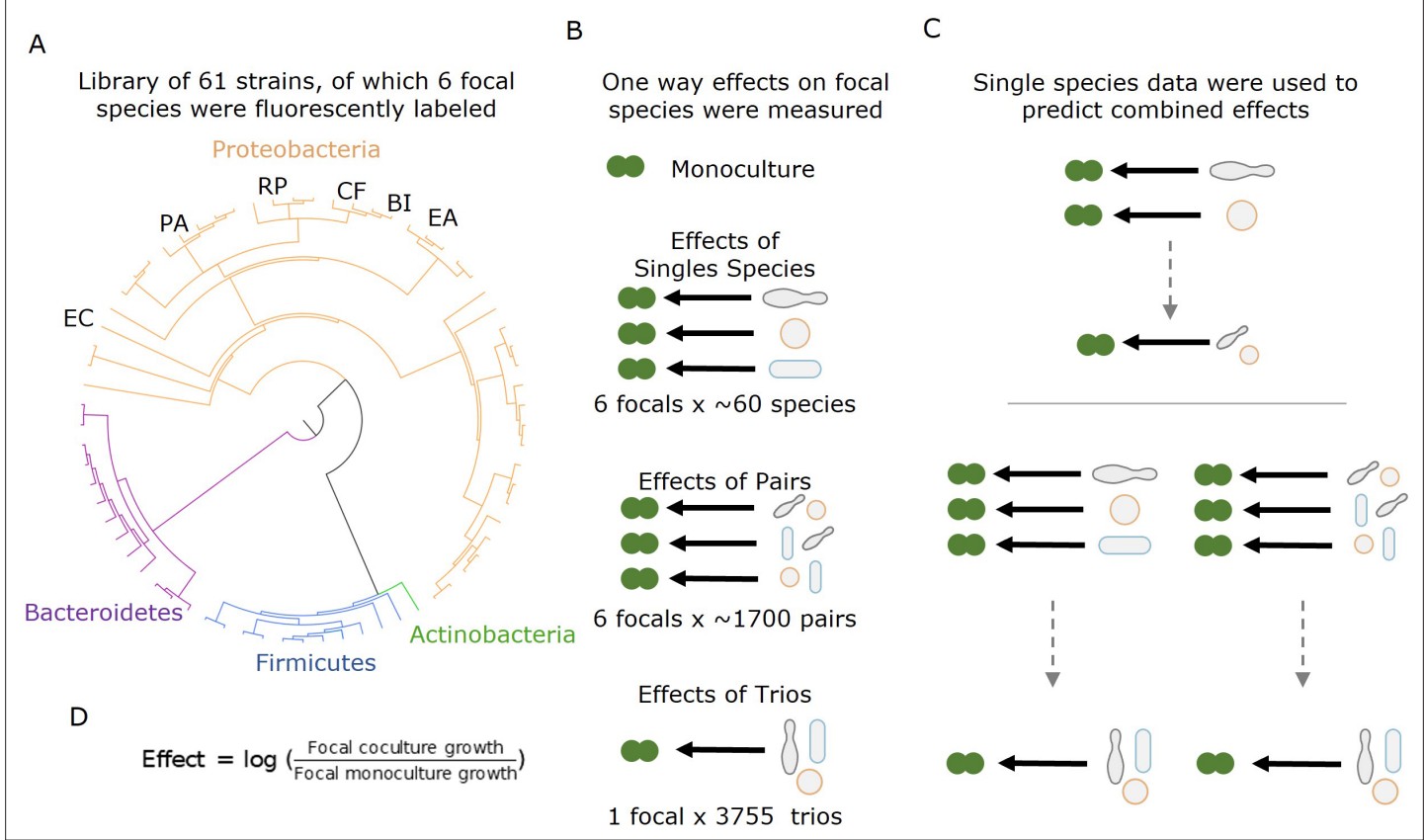

**Figure 1.** Measuring effects of 61 affecting species, and their pairs and trios on 6 focal species. (**A**) A library of 61 soil and leaf-associated bacterial strains was used in this experiment. All strains are from four orders: Proteobacteria (orange), Firmicutes (blue), Bacteroidetes (purple), and Actinobacteria (green) (full list in ***Supplementary file 1a***, ***Source data 1***). Also, 6 of the 61 species were labeled with GFP and used as 'focal' species whose growth was tested in the presence of the other isolates (affecting species). These strains are labeled on the phylogenetic tree (*Escherichia coli* [EC], *Ewingella americana* [EA], *Raoultella planticola* [RP], *Buttiauxella izardii* [BI], *Citrobacter freundii* [CF], and *Pantoea agglomerans* [PA].) (**B**) Each focal species was grown in monoculture, with (between 18 and 52) single affecting species, and (between 153 and 1464) pairs of affecting species. Additionally, *E. coli* was grown with 3009 trios of affecting species. (**C**) Effects of pairs and trios were then predicted using the effects of single species and single species and pairs, respectively. Predictions were made using three different models: additive, mean, and strongest (detailed in 'Results' and 'Materials and methods'). (**D**) Equation used for calculating the effect of an affecting species on the focal species.

The online version of this article includes the following figure supplement(s) for figure 1:

**Figure supplement 1.** Species in each droplet and well in kChip experiments.

**Figure supplement 2.** Effects on focal species are independent of initial species' density.

**Figure supplement 3.** Carbon source utilization profiles for bacterial strains.

**Figure supplement 4.** Antibiotic resistance profiles for bacterial strains.

**Figure supplement 5.** Growth dynamics of focal species in monoculture.

*2016*; *Venturelli et al., 2018*). This being the case, how microbial interactions combine into the joint effect of multiple species on a single species of interest is still poorly understood.

In our research, we used high-throughput nanodroplet-based microfluidics to measure over 14,000 bacterial communities composed of subsets of a library of 61 soil and leaf isolates of which six were fluorescently labeled (*Figure 1*). We quantified the effect of individual species and the joint effects of species pairs and trios on the growth of six focal bacterial species and found that the effects of multiple species are dominated by the strongest single-species effect, and specifically that negative effects combine non-additively.

## Results

We conducted high-throughput assays involving 61 affecting species and 6 different focal species to understand the effects of single species, pairs, and trios on the growth of a given (focal) species. The 61 affecting species included soil and leaf isolates as well as lab strains representing 19 genera from four phyla: Proteobacteria (n = 14), Firmicutes (n = 2), Bacteroidetes (n = 2), and Actinobacteria (n = 1) (full list in *Supplementary file 1*, *Source data 1*). The focal species were a subset of six of these species (all proteobacteria) that were transformed to constitutively express a fluorescent protein: (*Escherichia coli* [EC], *Ewingella americana* [EA], *Raoultella planticola* [RP], *Buttiauxella izardii* [BI], *Citrobacter freundii* [CF], and *Pantoea agglomerans* [PA]) (see 'Materials and methods'). Except for EC, which is a lab strain (*E. coli K-12* substr. *MG1655*), all focals were isolated from soil samples (*Kehe et al., 2021*). First, we characterized each species phylogenetically by performing Sanger sequencing of their 16S ribosomal RNA gene and phenotypically by growing each species on each of 20 different carbon sources and 11 antibiotics. The species showed large variability in carbon utilization profiles with no species growing well on all carbon sources (*Figure 1—figure supplement 3*). There was also high variability in growth on antibiotics with 15 species showing little or no growth on any antibiotics, while 16 species showed resistance to at least seven antibiotics (*Figure 1—figure supplement 4*).

We performed the interaction assays in the kChip microfluidics device (*Kulesa et al., 2018*; *Kehe et al., 2019*), allowing for extensive screening in parallel (see 'Materials and methods,' *Figure 1—figure supplement 1*). We measured the effects of 243 single species (18–52 for each of the six focal species), the joint effects of 5357 species pairs (between 153 and 1464 for each of the six focal species), and the joint effects of 3009 species trios (from a subset of 26 affecting species on one focal species). Cultures were normalized and mixed after pre-growth, such that the starting densities in the kChip were approximately 1:1 for all species in wells containing two droplets and two affecting species, but ratios varied in three droplet wells (see 'Materials and methods,' *Figure 1—figure supplements 1 and 2*). Interaction assays were carried out in minimal M9 media with 0.5% [w/v] glucose for 24 hr. The growth of the focal species was measured by fluorescence, and effects were calculated as the log ratio of growth in coculture to growth in monoculture (see 'Materials and methods,' *Figure 1D*). Positive and negative effects are defined as a net increase or decrease in growth compared to the monoculture respectively, while affecting species with no observable effect (see 'Materials and methods') were defined as neutral.

### Joint effects of species pairs tend to be stronger than those of individual affecting species

We started our interaction assays by measuring the individual effects of single affecting species on each of the focal species (see 'Materials and methods'). Individual effects covered a wide range (median = −0.15, interquartile range = 0.94) (*Figure 2A*), and positive effects (the focal species reaching a higher density in the presence of an affecting species than in monoculture) were common overall (32.9%, *Figure 2B*), in line with previous studies (*Kehe et al., 2021*). The distribution of effects varied based on the focal species, with *E. coli* and *B. izardii* showing the most negative (–2.83) and positive (+0.43) median effects, respectively (*Figure 2D*). Additionally, we found no affecting species that had strong effects across all focal species (*Figure 2—figure supplement 1*).

The measured traits of individual species showed no consistent correlations with their effects on the focal species. In particular, the similarity of metabolic profile, resistance profile, or phylogeny between the focal and affecting species did not correlate strongly with the effect across focal species. Some traits showed little to no correlation for most focals (e.g., antibiotic resistance), while other traits were correlated with effect for a number focal species but not all (e.g., phylogenetic distance). Most of these correlations were not statistically significant (*Figure 3—figure supplement 1A*).

After characterizing the individual effects of all single species, we assayed each pair of affecting species against the focal species. Overall, negative effects were significantly more prevalent in joint pair effects (77.1%) than in effects of individual species (60.5%) (p=1.3e-9, Fisher's exact test) (*Figure 2B and C*). The median effect on each focal was more negative by 0.28 on average, though the difference was not significant in all cases; additionally, focals with mostly positive single-species interactions showed a small increase in median effect (*Figure 2D*). Despite this, the minimum and maximum effects for each focal species remained similar. As with the single affecting species, pairs' joint effects did not correlate well with species traits, with similarity between the two affecting species,

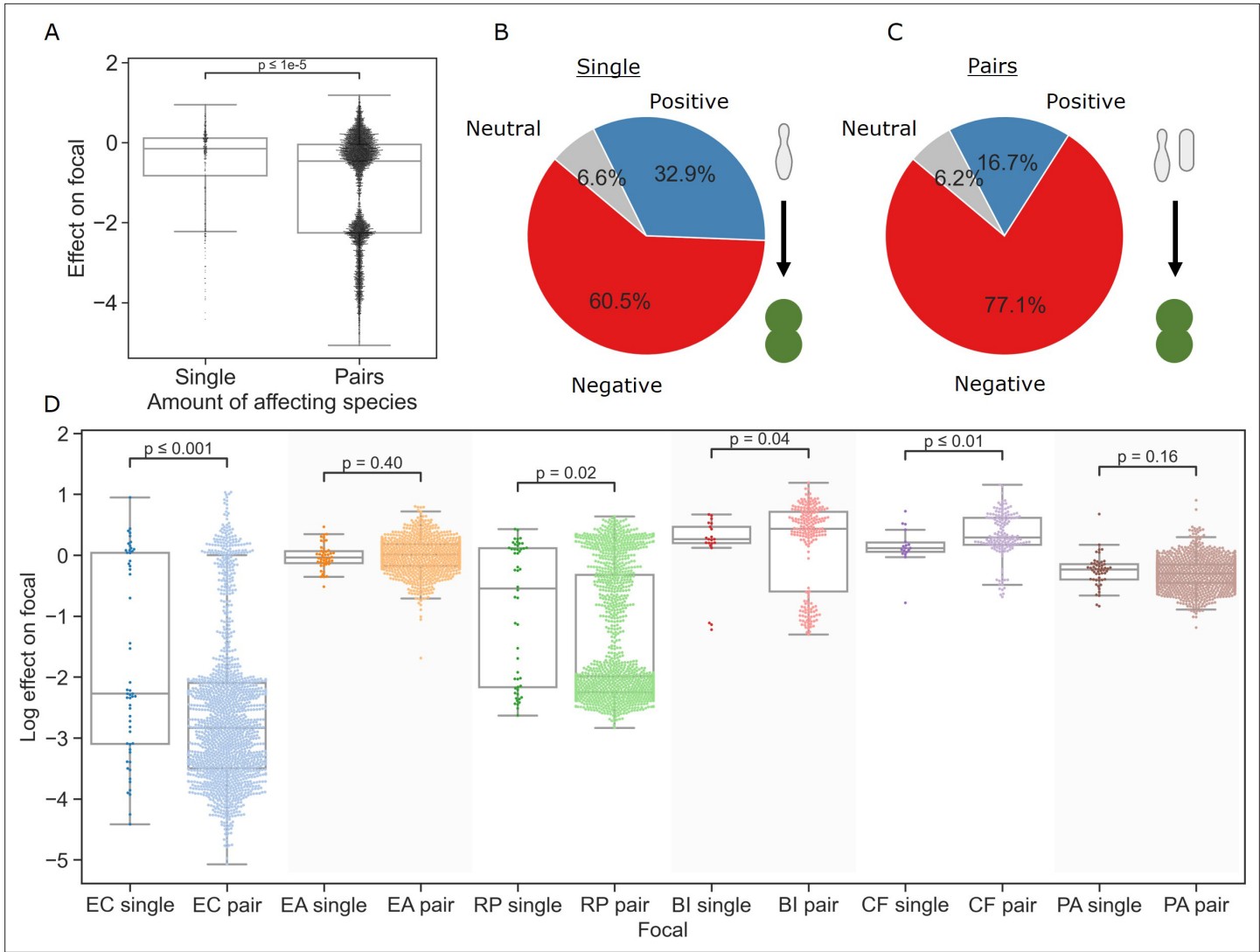

**Figure 2.** Pairs of affecting species have stronger effects than single species. (**A**) Distribution of the effects of single and pairs of affecting species on all focal species. Mann–Whitney–Wilcoxon test two-sided, p-value = 1e-9. Dots show individual effects, solid lines represent the median, boxes represent the interquartile range, and whiskers are expanded to include values no further than 1.5× interquartile range. (**B, C**) Distribution of qualitative effects of single and pairs of affecting species respectively on all focal species. (**D**) Distribution of the effect of single and pairs of affecting species for each focal species individually. Dots represent individual measurements, solid lines represent the median, boxes represent the interquartile range, and whiskers are expanded to include values no further than 1.5× interquartile range. Mann–Whitney–Wilcoxon test two-sided tests were performed for each focal species, and p-values are shown on the graph.

The online version of this article includes the following figure supplement(s) for figure 2:

**Figure supplement 1.** Minimal effect of single strains and pairs across multiple focals.

or with their similarity to the focal species (*Figure 3—figure supplement 1B*). These results indicate that it may be challenging to connect the effects of single and pairs of species on a focal strain to a specific trait of the involved strains using simple analysis.

## Negative effects combine non-additively and joint effects are dominated by the stronger single-species effect

Next, we examined how the effects of individual species relate to their joint effect. In particular, we were interested in finding a model that describes the effects of pairs based on the data from single-species effects. Based on previous studies' success in predicting community structure from pairwise interactions (*Friedman et al., 2017*; *Meroz et al., 2021*; *Guo and Boedicker, 2016*; *Venturelli et al.,*

*2018*), we posited that predicting how effects combine based solely on the effects of the single species should also be feasible. To do so, we considered three models: an additive effect model, a mean effect model, and a strongest effect model.

The additive effect model proposes that the effects of each species on the focal are the same whether they act individually or jointly. Therefore, the combined effect will be equal to the sum of the effects of each species on their own. This is equivalent to additivity of effects between antibiotics, which is common in the drug combinations field (*Bollenbach, 2015*). The mean model represents a simple phenomenological model that assumes that the effects of different species will be diluted in the presence of a third species. By contrast, the strongest effect model posits that the species with the strongest effect dominates the effect of other affecting species, leaving the joint effect the maximum single effect, and not the sum or mean of single-species effects.

When measured across all species and interaction types, we found that the model that best agrees with the measured effects is the strongest effect model (*Figure 3B*).Though supplementing the mean model with additional species information (i.e., carrying capacity) did improve the model accuracy, it was still less accurate than the strongest effect model (*Figure 3—figure supplement 2*). The accuracy of the models and identity of the best-fitting model varied across interaction types. The strongest effect model was the most accurate overall (nRMSE = 0.46, 0.32, and 0.16 for the additive, mean, and strongest models, correspondingly), and considerably more accurate when both species affected the focal negatively (nRMSE = 0.65, 0.25, and 0.16). The additive model was slightly more accurate when one effect was negative and the other positive (nRMSE = 0.14, 0.43, and 0.16). Overall, predictions when both effects were positive were less accurate, but here too the strongest model gave the most accurate predictions (nRMSE = 0.81, 0.78, and 0.69) (*Figure 3C*).

The distribution of errors further supported the strongest effect model (*Figure 3B*, *Figure 3—figure supplement 3B*): When both single-species effects were negative, the mean model was prone to underestimating the combined effect due to the reduction of the stronger effect by taking into account the weaker effect; while contrastingly, the additive model overestimated effects due to the addition of the weaker effect to the stronger effect, which was more accurate on its own. We saw the opposite trend when both single-species effects were positive, and no particular trend when there was one positive and negative effect. As with the effects themselves, model accuracy was not strongly correlated with any specific species trait (*Figure 3—figure supplement 4*).

In regard to negative effects, support for the strongest model is also evident in how the difference in size of effect influences the model accuracy (*Figure 3—figure supplement 3A*). When effects are close to equal, the mean model is fairly accurate while the additive model does particularly poorly as these effects would be calculated as twice the strongest effect. Contrastly, when one effect is much stronger than the other, the additive model is accurate since the addition of the weak effect is negligible, whereas the mean model underestimates the joint effects by taking into account the weaker effect.

## The strongest effect model is also the most accurate for larger communities

With this information in hand, we were interested to see whether the same rules held up for larger communities. To this end, we screened trios of a subset (i.e., 26) of the affecting species against a single focal species (*E. coli*) and found similar trends to all those seen for pairs of affecting species. Similar to what was observed in the move from single species to pairs, effects were stronger (in this case more negative effects) in trios than in the pairs (*Figure 4C*). Additionally, as with joint pairs' effects, the strongest effect model was more accurate than the additive and mean models (nRMSE = 2.65, 1.23, and 0.63 for the additive, mean, and strongest models, correspondingly), which is consistent with the fact that the single-species effects in this subset were predominantly negative. Similar distributions of error were seen as in the pairs' effects, but further exaggerated with the more extreme under and overestimation of the combined trios' effects by the mean and additive models, respectively (*Figure 4A*).

We further explored the additive, mean, and strongest models in trios by basing the model on effects of the three pairs comprising each trio (i.e., joint effect of AB, AC, BC to predict effect of ABC), as opposed to only using single-effect data (i.e. ,effect of A, B, C on their own) (*Figure 4A and B*). The effects of single species and pairs were measured again independently in this experiment

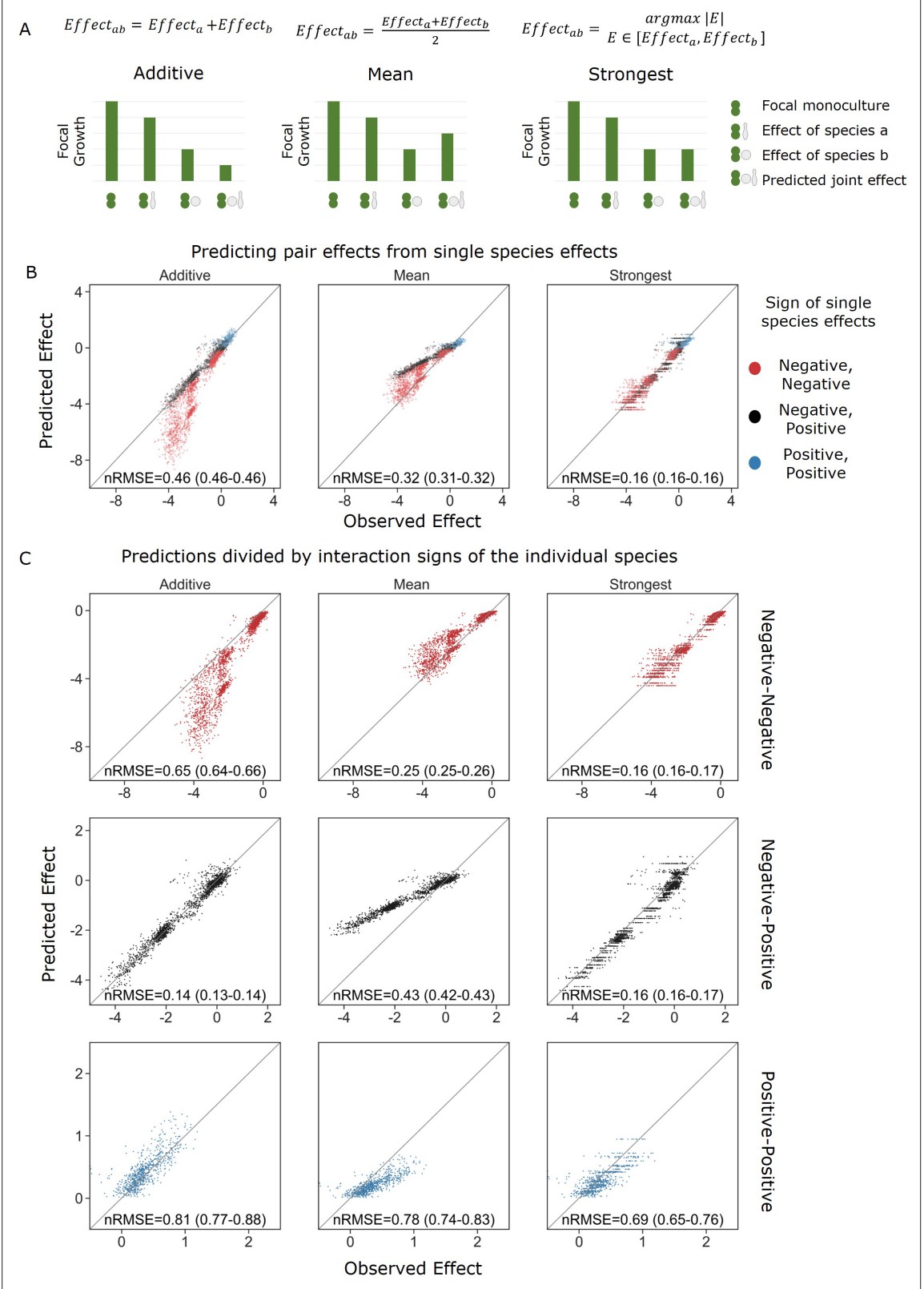

**Figure 3.** Strongest single-species effect offers the most accurate model for the combined effect of two species. (**A**) Graphical representation for each model. The additive model assumes that the effects of each species will accumulate, indicating they are acting independently, and are unaffected by one another. The mean model assumes the combined effect will be an average of the two single-species effects. The final model, strongest effect, assumes that whichever species had a stronger effect on its own will determine the joint effect when paired with an additional species. The y-axis

*Figure 3 continued on next page*

*Figure 3 continued*

represents the growth of the focal species in different conditions, and in these examples effects are negative. (**B**) Comparison of predicted effects and the experimental data, with their respective root mean squared error normalized to the interquartile range of the observed data (nRMSE). nRMSE values are calculated from 1000 bootstrapped datasets and represent the median and interquartile range in parentheses (see 'Materials and methods'). Each dot represents the joint effect of a pair of affecting species on a focal species. Colors indicate the signs of the measured effects of the individual affecting species. (**C**) Similar to panel (**B**), but data is stratified by interaction signs of the individual affecting species.

The online version of this article includes the following figure supplement(s) for figure 3:

**Figure supplement 1.** Correlation between affecting species traits and effect on focal.

**Figure supplement 2.** OD-weighted mean model.

**Figure supplement 3.** Distribution of errors for each model predicting pair effects from single species.

**Figure supplement 4.** Traits effect on model error.

**Figure supplement 5.** Accuracy of all models is reduced when considering only combinations of strains that have weak effects.

**Figure supplement 6.** Model comparisons stratified by focal species and interaction type.

(see 'Materials and methods,' *Figure 1—figure supplement 1*). The move to pairs-based predictions improved the accuracy for both the mean and strongest model, while further pushing the additive predictions away from the observed effects (*Figure 4B*). These data suggest that even in the presence of additional species the strongest single-species effect still dominates the combined effect of a community.

## Discussion

By measuring thousands of simplified microbial communities, we quantified the effects of single species, pairs, and trios on multiple focal species. The most accurate model, overall and specifically when both single-species effects were negative, was the strongest effect model. This is in stark contrast to models often used in antibiotic compound combinations, despite most effects being negative, where additivity is often the default model (*Bollenbach, 2015*). The additive model performed well for mixed effects (i.e., one negative and one positive), but only slightly better than the strongest model, and poorly when both species had effects of the same sign. When both single-species effects were positive, the strongest model was also the best, though the difference was less pronounced and all models performed worse for these interactions. This may be due to the small effect size seen with positive effects, as when we limited negative and mixed effects to a similar range of effects strength, their accuracy dropped to similar values (*Figure 3—figure supplement 5*). We posit that the difference in accuracy across species is affected mainly by the effect type dominating different focal species' interactions, rather than by inherent species traits (*Figure 3—figure supplement 6*).

We phenotypically and genetically profiled all species, but did not find strong correlations between the measured traits, or similarity in traits, to the effect on the focal species. Though positive effects were common, making up about one-third of the single-species effects, they became less common as the number of community members increased, making up only 16% of the effects of species pairs. Furthermore, we found similar trends in the larger communities of four species (three affecting species and one focal), both that effects combined in a non-additive manner, being dominated by the strongest single-species effect, and that effects became stronger in larger communities, which is consistent with previous studies (*Cook et al., 2006*; *van Elsas et al., 2012*; *Jones et al., 2021*; *Piccardi et al., 2019*; *Gould et al., 2018*; *Palmer and Foster, 2022*).

The mechanistic basis underlying the joint effects of multiple species is still unclear. The additive model's accuracy for mixed effects may indicate that negative and positive effects act independently. For negative effects, it is difficult to identify a single biological mechanism that could explain why the strongest effect model agreed best with our experimental data. Intuitively, we assumed this could be explained by resource competition (i.e., an affecting species that consumes resources quickly would negatively affect the focal species, as well as the other affecting species). However, this explanation is not consistent with the fact that the affecting species' growth rate did not correlate well with their effect on some focal species (*Figure 3—figure supplement 1A*). Secondly, we thought effects could be saturating (either biologically or with regard to the detection limits in this experimental setup), but this would not explain why the model works for weaker effects. A hierarchical ranking, where each

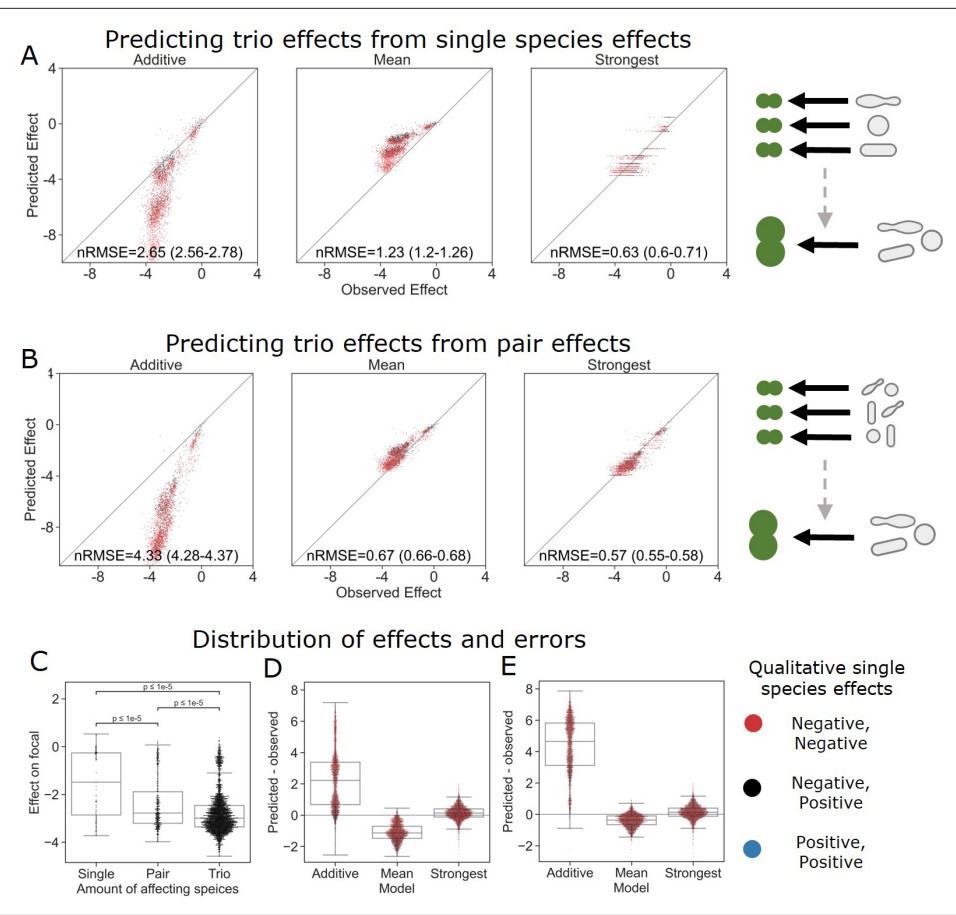

**Figure 4.** The strongest effect model is also the most accurate for trios. (**A, B**) Correlation between three different models for how (**A**) single-species effects and (**B**) pairwise species effects combine into trio effects, and the experimental data. Root squared mean error normalized (nRMSE) to the interquartile range. nRMSE values are calculated from 1000 datasets and represent the median and interquartile range in parentheses (see 'Materials and methods'). (**C**) Distribution of the effects of single, pairs, and trios of affecting species on *E. coli*. All Mann–Whitney–Wilcoxon two-sided tests were significant, p values are shown on plot. Dots show individual effects, solid lines represent the median, boxes represent the interquartile range, and whiskers are expanded to include values no further than 1.5× interquartile range. (**D, E**) Distribution of errors for each model based on (**D**) single-species data and (**E**) pairs data. Dots show individual effects, solid lines represent the median, boxes represent the interquartile range, and whiskers are expanded to include values no further than 1.5× interquartile range.

The online version of this article includes the following figure supplement(s) for figure 4:

**Figure supplement 1.** OD-weighted mean model.

species affects all the species ranked below it could lead to the strongest affecting species affecting both the focal and the other affecting species, thus dominating their joint effects, but this does not coincide with the fact that we observed almost no single species or pairs with a strong effect across all focals (*Figure 2—figure supplement 1*).

As we did not measure the abundance of all species in each community (only the focal), we cannot disentangle interaction modification (changes in per capita effect of specific species), from interaction chains (affecting the amount of an affecting species, and as such its effect on the focal), and further work is needed in order to pinpoint the exact mechanism(s) leading to the dominance of the strongest model for negative effects in our system. We also note that it is possible that the manner in which effects combine is affected by the mechanism of interaction; For instance, previous studies have shown that interference competition can combine additively, or even synergistically, results not seen in our work (*Tyc et al., 2014*; *Westhoff et al., 2021*).

Understanding how microbial communities assemble and how large numbers of species interact is of both utmost importance and difficulty. Harnessing such information would open up a plethora of currently underutilized applications in food, medical, and agricultural industries. Specifically, understanding how the effects of multiple species on a single species combine is important for introduction of new species into a given environment. Our results suggest that when we want to affect a single focal species in a given environment (e.g., for biocontrol of a pathogen), introducing the species with the strongest effect on the focal would be sufficient to obtain the desired effect, as synergies were rare in our dataset. In cases where there are multiple strains of interest (e.g., probiotics), introducing multiple species may be beneficial since different affecting species typically have strong effects on different focals. Introducing combinations of species may allow for a more robust function as the chances that one member of the community will have a strong effect on a resident species of interest is more likely.

Further work is needed in order to deepen our understanding of how multiple species affect each other and to see to what extent our findings continue to hold up in more diverse communities, other taxonomic groups, and more complex environments. Specifically, as we saw a decrease in prediction power from pairs to trios, exploring this model with more diverse communities is of particular interest. Additionally, nearly all of the effects in the four member communities were negative, and it is unclear how mixed and positive effect modeling is affected by higher diversity. Lastly, it is important to note that our focal species are all from the same order (Enterobacterales), which may also limit the purview of our findings. Nonetheless, our results suggest that community effects can be predicted from the strongest effect of a single species, greatly reducing the amount of information required to obtain accurate estimations, which can improve our ability to use a bottom-up approach for biotechnological applications, as well as answering fundamental ecology questions.

## Materials and methods
### Strain isolation from soil samples
Soil (50 ml of soil, taken from a depth of ~30 cm) and leaf samples (multiple leaves from a single plant combined into a sterile 50 ml tube) were collected from various locations in the Faculty of Agriculture in Rehovot, Israel, on multiple dates (See *Source data 1* file for more information). Each sample was diluted in phosphate-buffered saline (PBS) directly after collection (1 g of soil or one leaf in 10 ml of PBS) and vortexed for 5 min. 100 µl of multiple dilutions of this mixture ($10^{-2}$-$10^{-5}$) were seeded on different solid media NB (0.5% [w/v] peptone, 0.3% [w/v] yeast extract, 1.5% [w/v] agar); 1% NB (0.005% [w/v] peptone, 0.003%[w/v] yeast extract, 1.5% agar); M9 minimal media (0.1 mM $CaCl_2$, 2 mM $MgSO_4$, 1× [Enco-teknova] trace metals, 1% [w/v] glucose, 1× [Sigma] M9 Salts), additional plates were made with the same media containing various antibiotics (antibiotics and respective concentrations are listed in *Supplementary file 1b*) Plates were incubated at 30°C, and colonies were restreaked on NB without antibiotics until single isolates were stably obtained. Strains were selected on the basis of multiple criteria: growth of transferred colony in NB liquid medium (30°C), frozen glycerol stock revival in NB ($OD_{600} > 0.1$) (30°C), and subsequent growth on M9 minimal media + 1% (w/v) glucose ($OD_{600} > 0.1$) (30°C). Isolates were kept in single tubes as well as 96-well plates in 50% NB + 50% glycerol (glycerol stock were 60% and 80% for tubes and plates, respectively, for 30% and 40% final glycerol concentrations).

### Strain identification and phylogenetic distance calculation
Each bacterial isolate was classified phylogenetically with its 16S rRNA gene sequence. The full 16S gene sequences (~1500 base pairs) were obtained via Sanger sequencing and classified with a combination of RDP Classifier (*Cole et al., 2014*) and BLAST (*Altschul et al., 1990*) (list of strains in *Supplementary file 1*, full phylogenetic data in *Source data 1*). Phylogenetic distance was calculated in Geneious Prime software (version 2022.2.1, Biomatters Ltd). Sequences were aligned using MUSCLE alignment. Phylogenetic tree was built using the UPGMA method with no outgroup and a HKY genetic distance model. The pairwise phylogenetic distances between strains were calculated directly from the patristic distances of the phylogenetic tree.

## Phenotypic profiling and distance calculation

Bacterial strains were seeded from –80 stock directly into 1 ml LB medium (1% [w/v] tryptone, 1% [w/v] NaCl, 0.5% [w/v] yeast extract) in 96-well plate and grown overnight at 30°C at 900 RPM (on a Titramax 100; Heidolph Instruments, Schwabach, Germany). Cells were washed three times by centrifugation as 3600 rcf, removal of supernatant, and resuspension in M9 minimal media with no carbon. Cultures were then normalized to 0.01 $OD_{600}$. 20 µl of the normalized cultures were added to 180 µl of M9 minimal media either containing 1% [w/v] of one of 20 carbon sources (*Supplementary file 1*) or M9 minimal media with 1% [w/v] glucose and one of 11 antibiotics (antibiotics and respective concentrations are listed in *Supplementary file 1*). Plates were grown at 30°C for 48 hr without shaking. Cultures were homogenized by shaking (on a Titramax 100) for 90 s before measuring $OD_{600}$. Additionally, species were also grown in M9 minimal media (with the addition of 0.05% [w/v] BSA and 1% [w/v] glucose) with shaking (continuous double orbital shaking, 282 cpm) at 30°C, with OD measurements every 15 min, for 48 hr, to obtain growth kinetics (in Epoch and Synergy H1 microplate readers). Growth for carbon source experiments was normalized to the carbon source with the highest $OD_{600}$ and antibiotic experiments were normalized to growth on M9 minimal media with glucose and no antibiotics. Euclidean distances of normalized values were measured for each species on carbon sources and antibiotics separately, and used to construct distance matrices. Growth kinetics (i.e., growth rate and carrying capacity) were not included in these profiles, but measured independently for correlation to effect size.

## Droplet preparation and culturing

Bacterial strains were seeded from –80 stock directly into 1 ml LB medium (1% [w/v] tryptone, 1% [w/v] NaCl, 0.5% [w/v] yeast extract) in 96-well plate and grown overnight at 30°C at 900 RPM (on a Titramax 100). Cells were washed three times by centrifugation as 3600 rcf for 3 min, removal of supernatant, and resuspension in M9 minimal media (with the addition of 0.05% [w/v] BSA and 1% [w/v] glucose). Affecting species cultures were then normalized to 0.04 $OD_{600}$ and focal species cultures were to 0.02 $OD_{600}$. Affecting and focal species cultures were combined at a ratio of 1:1 so that droplets contained a final concentration of 0.02 $OD_{600}$ affecting species and 0.01 $OD_{600}$ focal species. Each well contained droplets with the same focal species such that with this setup, in a well containing two droplets of different affecting species, the starting $OD_{600}$ of each species is 0.01 (as each affecting species is diluted by the other droplet in which it is not contained, but the focal species is not). In wells with three droplets, the starting ratio of the focal to each affecting species (assuming different species in each droplet) was 3:2. When one of the droplets contains a monoculture of the focal or is empty, or more than one droplet contains the same affecting species, these ratios change (see *Figure 1—figure supplement 1*).

Droplets were produced on a Bio-Rad QX200 Droplet Generator as described by *Kulesa et al., 2018* Briefly, 20 µl input of combined cultures were emulsified into ~20,000 1 nl droplets in fluorocarbon oil (3M Novec 7500) stabilized with 2% (w/w) fluorosurfactant (RAN Biotech 008 FluoroSurfactant). 2.5 mM of fluorescent dyes (Thermo Fisher AlexaFluor: 555 [A33080], 594 [A33082], 647 [A33084]) were added to culture for droplet imaging (see *Kulesa et al., 2018*). For each kChip loading, about 5000 droplets for each input (~60 affecting species + focal species, 2 focal species monocultures, 2 blank cultures) were generated for a total of ~320,000 droplets. Droplets were generated together for 2 kChips (technical replicates), and then droplets were pooled separately for each chip. kChips were incubated at 30°C for 72 hr. Cultures were imaged at 24 hr intervals throughout the experiment. Data for analysis was taken from after 24 hr as monoculture growth of the focals saturated by this point (*Figure 1—figure supplement 5*).

## Fluorescence labeling and assays

Focal species were transformed with commercially available plasmid pMRE132 containing GFP2 by *Kehe et al., 2021*. Fluorescence has some caveats as a measurement for biomass, as fluorescent signal is not always directly proportional to biomass, expression levels can vary in different physiological states, and signal stability can differ between strains. Nonetheless, as described in Appendix 1, we show that effect sizes assayed using fluorescence and standard $OD_{600}$ are well correlated (*Appendix 1—figure 1*).

## Data filtering and normalization

As the kChip genreates droplet combinations stochastically, the amount of replicates for each community is different, ranging from 1 to 285, with a mean of 19. All communities with less than three replicates were not used in the analysis. Additionally, isolates were only used with focals whose monocultures were at least five times larger than the isolates autofluorescence signal, allowing to measure effects of at least –1.5. Full datasets without autofluorescence filtering can be seen in *Appendix 1—figure 3*. Importantly, affecting species autofluorescence would weaken measured negative effects and would not systematically generate artifacts that support the strongest effect model. Normalization was performed by subtracting the starting value for each individual well from the additional time points.

## Calculating effect size

To measure the effect of each affecting species on a given focal species, the log of the ratio of focal species yield in coculture (median of coculture replicates) to monoculture (median of monoculture replicates) was calculated:

$$Effect_i = log(\frac{Median\ growth\ of\ focal\ in\ coculture\ with\ species\ i}{Median\ growth\ of\ focal\ in\ monoculture}).$$

Coculture data was collected from wells with different starting concentrations in both the two-droplet experiments (i.e., one mixed droplet and one focal monoculture) and three-droplet experiments (i.e., two droplets of affecting species A and one droplet of affecting species B and vice versa, or one of each in addition to a blank droplet) (*Figure 1—figure supplement 1*). Our data showed that the different initial fractions did not influence the effect on the focal species (*Figure 1—figure supplement 2*). The standard error was calculated via bootstrapping, 100 calculations of the resampled median coculture divided by resampled median monoculture. Effects where the standard deviation was larger than the absolute value of the effect were classified as neutral.

## Calculating predictions for different models and their accuracy

The additive model assumes that the effects of each species will accumulate, and is the combined effect is the sum of effects, calculated as

$$Effect_{(1...n)} = Effect_1 + ... + Effect_n$$

The mean model assumes the combined effect will be an average of the two single species effects and is calculated as

$$Effect_{(1...n)} = \frac{Effect_1 + ... + Effect_n}{n}$$

The OD-weighted mean model weighs the mean of effects by the affecting species' maximum $OD_{600}$ in the growth curves experiment and is calculated as

$$Effect_{(1...n)} = \frac{Effect_1 * max_{OD_{600}1} + ... + Effect_n * max_{OD_{600}n}}{max_{OD_{600}1} + ... + *max_{OD_{600}n}}$$

The strongest effect model assumes that whichever species had a stronger effect on its own will determine the joint effect when paired with an additional species. It is calculated as

$$Effect_{1...n} = \begin{array}{c} argmax\ |E| \\ E \in [Effect_1, ..., Effect_n] \end{array}$$

returning the effect with the largest absolute value (e.g., if two single species' effects are –3 and +1, the model will predict that their joint effect is –3).

Root mean square error measuring the accuracy of each model was normalized to the interquartile range for each dataset. Normalized root mean square error median and interquartile ranges were calculated via bootstrapping. The dataset from each focal was sampled 1000 times with replacement. Sampling was done for individual effect measurements (specific wells), and median effect sizes for

species, pairs, and trios were recalculated from these sampled datasets. The sampled datasets from each focal were assembled into 'full' datasets (containing all focals) from which nRMSEs were calculated. The median and interquartile range of the normalized root mean square errors were calculated from the 1000 sampled datasets' values.

## Acknowledgements

We thank Jared Kehe and Anthony Ortiz for helping set up the kChip system in our lab, as well as Meilin Zhu, Megan Tse, Julie Chen, and Paul Blainey for continued correspondence and technical support. Alfonso Pérez Escudero, and members of the Friedman lab for helpful discussions. Nadav Kashtan for constructive comments on the manuscript. Eddie Cytryn for generously providing bacterial strains used in this study. Lastly, Yael Sorokin for technical assistance. This research was supported by the United States–Israel Binational Science Foundation (grant no. 2017179).

## Additional information

### Funding

| Funder | Grant reference number | Author |
|---|---|---|
| United States - Israel Binational Science Foundation | 2017179 | Amichai Baichman-Kass Tingting Song Jonathan Friedman |

The funders had no role in study design, data collection and interpretation, or the decision to submit the work for publication.

### Author contributions

Amichai Baichman-Kass, Conceptualization, Data curation, Formal analysis, Writing - original draft, Writing - review and editing; Tingting Song, Isolated strains used in this study; Jonathan Friedman, Conceptualization, Supervision, Funding acquisition, Writing - original draft, Writing - review and editing

### Author ORCIDs

Amichai Baichman-Kass (ID) http://orcid.org/0000-0003-0924-3191
Tingting Song (ID) http://orcid.org/0000-0002-6740-2085
Jonathan Friedman (ID) http://orcid.org/0000-0001-8476-8030

### Decision letter and Author response

Decision letter https://doi.org/10.7554/eLife.83398.sa1
Author response https://doi.org/10.7554/eLife.83398.sa2

## Additional files

### Supplementary files

• MDAR checklist

• Source data 1. This file contains the following information regarding the strains in this study: Phylum, Class, Order, Family, and Genus, Nearest species (assigned using BLAST), whether species was used in the trios experiment, Identifier (a taxonomic name for each strain based on the nearest species), 16s rRNA sequences, Genebank accession number for strains isolated in this study.

• Supplementary file 1. This file contains two supplementary tables: the strains used in this study (further details can be found in *Source data 1*) and the carbon sources and antibiotics (including concentrations) used for phenotypic characterization of the strains.

### Data availability

Sequencing data are provided in *Source data 1*, and have been deposited to Genbank under accession codes OP389073-OP389107 and OP412780-OP412788.All data generated or analyzed during this study are available on GitHub, and can be found at https://zenodo.org/badge/latestdoi/534114367.

The following dataset was generated:

| Author(s) | Year | Dataset title | Dataset URL | Database and Identifier |
|---|---|---|---|---|
| Friedmann J, Baichman-Kass A, Song T | 2022 | amichaibk/community_ effects: Interactions between culturable bacteria are highly non-additive data and analysis code | https://doi.org/10.5281/zenodo.7101221 | Zenodo, 10.5281/ zenodo.7101221 |

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

## Appendix 1

### Fluorescence assays

To test the accuracy of using fluorescence to assay interactions, we performed the following experiment correlating effect size as measured by fluorescent signal to effect size as measured by $OD_{600}$. Bacterial strains were seeded from –80 stock directly into 0.5 ml LB medium in a 96-well plate, and grown overnight at 30°C at 900 RPM (on a Titramax 100). Cells were washed three times by centrifugation at 3600 rcf for 3 min, removal of supernatant, and resuspension in M9 minimal media (with the addition of 1% [w/v] glucose). All cultures were normalized to 0.02 $OD_{600}$. HTD96b plates (HTDialysis, Gales Ferry, CT) with membranes containing 1 μm pores splitting each well were seeded with 150 μl affecting species and focal species cultures on opposite sides of the membrane. After a 24 hr growth period at 30°C, shaking at 600 RPM, 100 μl of culture for each side of each well was transferred to a standard 96-well plate and $OD_{600}$ and fluorescence were measured (*Appendix 1—figure 1*). Each interaction was measured using three technical replicates.

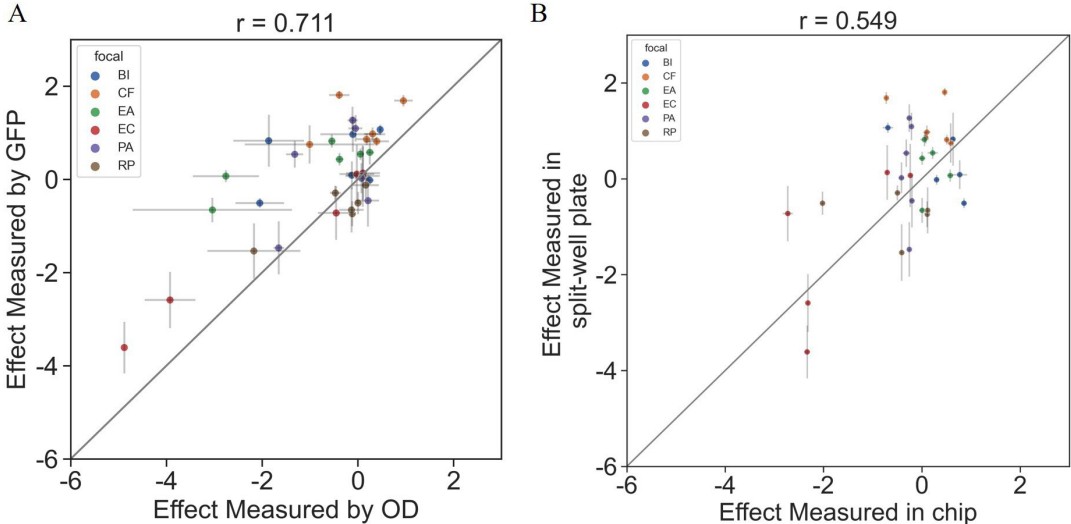

**Appendix 1—figure 1.** Interactions measured with $OD_{600}$ are consistent with those based on fluorescent measurements. The effects of six single species on each focal were measured using $OD_{600}$ and fluorescence in a 96 split well plate (see 'Materials and methods'). (**A**) The correlation between the effect when measured by fluorescence and $OD_{600}$ (p=1e-6). (**B**) The correlation between the effect when measured in the kChip and the HTD Equilibrium Dialysis System (p=0.001). Each point represents the median effect of three techinacal repliactes, and error bars represent the standard error calculated viabootstraping (see 'Materials and methods').

To ensure that model accuracy was not influenced by (fluorescent) measurement limitations, we analyzed the competitive effects of models with predictions limited to the range of minimal observed measurements (as we know the maximal measurements were not near saturation). This affected only the additive model (which was the only model that could predict effects stronger than those observed), and its accuracy was improved, but it was still less accurate than the mean and strongest models (*Appendix 1—figure 2*).

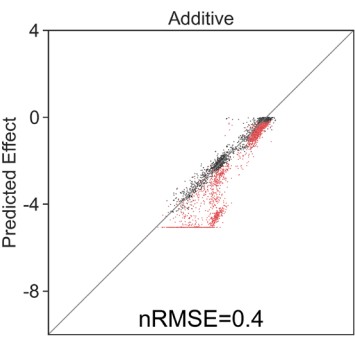 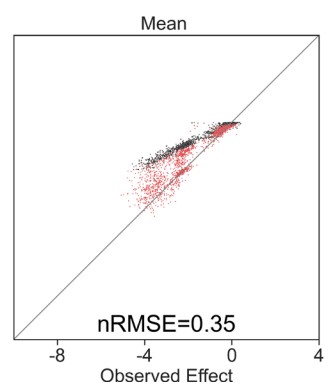 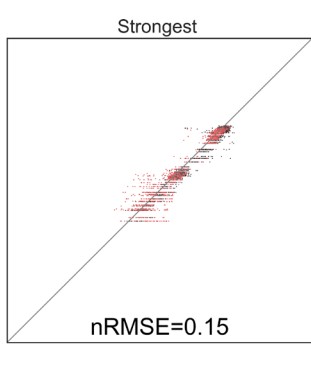

**Appendix 1—figure 2.** Model predictions limited by the lowest observed effect. Correlation between three different models for how single species effects combine, and the experimental data, with their respective normalized root squared mean error. Model predictions were limited to the minimal observed effects, and only data for negative predictions are shown.

Additionally, an experiment was carried out in the kChip to measure autofluorescence of affecting species. This setup was identical to the droplet preparation and culturing protocol detailed above, except that cultures were not mixed with the focal species prior to droplet generation. In this setup, each droplet contains a single species, and wells contain one or two species (depending on whether the droplets were from the same or different cultures). Isolates were only used with focals whose monocultures were at least five times larger than the isolates autofluorescence signal, allowing to measure effects of at least –1.5. Full datasets without autofluorescence filtering can be seen in *Appendix 1—figure 3*.

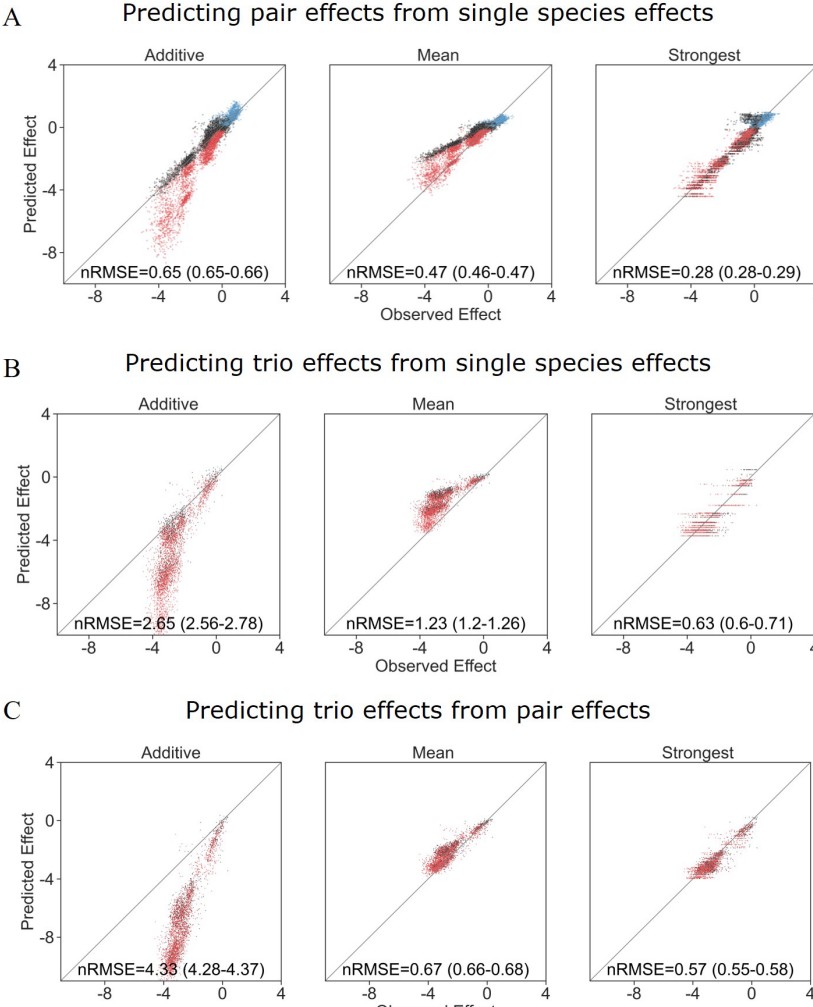

**Appendix 1—figure 3.** Different models prediction accuracy using all measured effects, not filtered for affecting species autofluorescence. Correlation between three different models for how (**A**) single-species effects combine to pairs, (**B**) single-species effects combine into trio effects, and (**C**) pairwise species effects combine into trio effects, and the experimental data. Root squared mean error normalized (nRMSE) to the interquartile range. nRMSE values are calculated from 1000 datasets and represent the median and interquartile range in parentheses (see 'Materials and methods').

