## [Editor Report]

This important study presents an interesting example of how complexities of communities may be reduced by showing that when partner species exert a negative effect on the focal species, the joint effects are generally not additive, but rather dominated by the strongest single effect. The evidence, enabled by thousands of measurements using nanodroplet-based microfluidics, is compelling, although the generality of the conclusion awaits further studies. This paper is of interest to microbial ecologists and synthetic biologists.

---

## [Decision Letter]

**Decision letter after peer review:**

Thank you for submitting your article "Interactions between culturable bacteria are highly non-additive" for consideration by *eLife*. Your article has been reviewed by 3 peer reviewers, including Wenying Shou as the Reviewing Editor and Reviewer #1, and the evaluation has been overseen by Meredith Schuman as the Senior Editor. The following individual involved in review of your submission has agreed to reveal their identity: Alvaro Sanchez (Reviewer #2).

Essential revisions:

1) Revise the text with a softer tone. For example, the generality of the conclusions is unknown since the 6 focal species were from 1 family, and since the conclusion only works for a subset of focal species, etc. Please see specific comments especially from Reviewers 2 and 3 about revising the claims.

2) Provide a more rigorous analysis of the fluorescence assay. Exactly what is this assay measuring? How might the correlation between growth/biomass and fluorescence be affected by death, diauxic shift, or prolonged lag / stationary phases? See comments to the authors from Reviewer 3.

*Reviewer #1 (Recommendations for the authors):*

My only suggestions are on writing. Certain key aspects of experiments can be outlined in the main text. For example, when pairs were introduced, did the inoculum for each strain get halved? Figure 3A graphic is unclear, and can be made much clearer by plotting mono effects and predicted coeffects under the three models.

*Reviewer #2 (Recommendations for the authors):*

Figure 3: (A) I think this panel could be made clearer. What does the green bar represent? It is not explicitly clarified in the caption. I think it depicts the growth of the green "test" bacteria? Maybe clarifying this in the caption would help. Also, why not add the equation representing the interaction?

Figure 3: (B). The data points are so tiny it is difficult to see them. Maybe using larger dots (I understand there are a lot of them, though. I wonder if there is a clearer way to plot this data). Also the color choice is not the friendlies to color-challenged readers like this reviewer. I had trouble in particular distinguishing the pairs when one species had a positive while the other one had a negative effect from those that were both negative.

Lines 167-169: The authors introduce the "mean effect model". What is the theoretical justification for including this model in the analysis? I mean, in terms of ecological theory? The additive model is justified e.g. in that it is the typical assumption in Lotka-Volterra (and also considered in the antibiotic combination literature cited in the paper). But how about the mean model? It would help if the authors explained/justified the theoretical basis for this. Otherwise it feels a bit random, they could have taken the median, or the square root of the variance, or…

Lines 221-224: The authors write: "We further explored this model by basing trios' data not on the additive, mean, or strongest values of the effects of individual species, but on those of the joint effects of the three pairs comprising each trio (the effects of single species and pairs were measured independently again in this experiment, see Materials and methods, Figure S1)." I have read this sentence five times and I am still not sure what the authors meant

Lines 296-300. The authors may want to more explicitly bring up in their discussion that the predictive ability of the "Dominance", strongest-species model, is significantly worse when they try to predict the effect of a trio than when they try to predict the effect of a pair. This would suggest some caution in the reach of their conclusions, as it is possible that the predictive power of a single species will get worse and worse as the diversity of the community increases. Which, by the way, would not be surprising I think and should not be held against their findings, but I still think it would be good to qualify their statements about the potential reach of their conclusions for the bottom-up prediction of population-level interactions in complex communities. While they do state that "further work is needed…" to figure out if they results hold in more diverse communities (Lines 294-96) I felt that the limitations of the study could be written in a sharper manner.

Finally: I was curious if the authors have considered a model where one of the species is dominant in a pair, but the one that dominates is not necessarily the one with the strongest effect? For instance, is it possible that when A is grown with either B alone or C alone, the suppression of growth from B is stronger than the suppression of growth from C. Yet, in the presence of both B and C, the suppression of growth is exactly the same as that by C (or just closer to C than B)? Do the authors see this in any of their pairs? If so, how many?

*Reviewer #3 (Recommendations for the authors):*

The metric of growth for the focal species was fluorescence. This can be a risky measurement, because other species could autofluoresce in the emission spectrum. Additionally, fluorescent proteins can continue to fluoresce after cell death and lysis (we have personally observed this after phage infection and antibiotic treatment). I think the paper could use a test to verify that fluorescence was an unbiased proxy for growth.

I am confused by the densities that the species start at. In the methods, it says the focal and affecting species had starter cultures that were 2-fold different in concentration, yet were mixed 1:1, and ended up with a 1:1 ratio. How is this possible? Supp Figure S1 did not help me understand this.

It was surprising to me that inoculation density had no effects. This makes me wonder whether the interactions observed in this study are dominated by primary metabolic competition, because density effects are very common when allelopathy occurs. If this is true, it restricts the generality of the results, and is worth being discussed. Related to this, antibiotic resistance was measured, but what about potential to secrete antibiotics?

I think the density effects should be measured with nRMSE, or even absolute difference from y=x, because there could be a strong correlation without the actual numbers being the same sign or magnitude. For example, in S11-B, most of the datapoints appear below y=x, until the effects are near zero, suggesting an effect-size-specific effect of density.

More discussion could be given on what a "meaningful" difference in nRMSE is.

More details on how the resampling during the bootstrap procedure was done is warranted.

It isn't clear to me how Figure 1C is supposed to show that different models were used-perhaps drop this and just leave this explanation for Figure 3A, which is very clear.

Suggestion for additional analysis using the trait data: this dataset seems perfect for using something like a random forest or other continuous-response, "open box" machine learning approach to agnostically ask whether the trait measurements can be used to predict effect when all the measurements are used, rather than summaries of the measurements in the distance metrics.

---

## [Author Response]

Essential revisions:1) Revise the text with a softer tone. For example, the generality of the conclusions is unknown since the 6 focal species were from 1 family, and since the conclusion only works for a subset of focal species, etc. Please see specific comments especially from Reviewers 2 and 3 about revising the claims.

We have put much effort into better understanding our findings in a manner which allows for a more generalizable framework, while also clarifying the remaining caveats of our study. In regards to the model working best for a subset of the focal species, we believe this due not to inherent characteristics of the focal species, but rather the types of interactions measured for each species. The strongest model works best for pairs where both interactions are qualitatively similar (i.e. positive-positive or negative-negative). Mixed interactions (i.e. negative-positive) were predicted slightly better by the additive model, and overall the predictions were less accurate for positive interactions. As such, focal species with many positive single species effects incur less accurate results for the model overall, and are not always best predicted by the strongest model. To reflect this, we have made revisions throughout the manuscript, including to the title and abstract of the paper.

Furthermore, we have made extensive changes to the discussion, adding caveats regarding the phylogeny of our focals, shortcomings of our models' predictions, and overall softened the tone as suggested. Detailed descriptions of these changes can be found in our responses below. In addition to multiple changes throughout the manuscript, the final paragraph of the Discussion section summarizes many of these caveats as follows:

"Further work is needed in order to deepen our understanding of how multiple species affect each other and to see to what extent our findings continue to hold up in more diverse communities, other taxonomic groups, and more complex environments. Specifically, as we saw a decrease in prediction power from pairs to trios, exploring this model with more diverse communities is of particular interest. Additionally, nearly all of the effects in the 4 member communities were negative, and it is unclear how mixed and positive effect modeling is affected by higher diversity. Lastly, it is important to note that our focal species are all from the same order (Enterobacterales), which may also limit the purview of our findings. Nonetheless, our results suggest that community effects can be predicted from the strongest effect of a single species, greatly reducing the amount of information required to obtain accurate estimations, which can improve our ability to use a bottom-up approach for biotechnological applications, as well as answering fundamental ecology questions. "

(Lines 359-369)

2) Provide a more rigorous analysis of the fluorescence assay. Exactly what is this assay measuring? How might the correlation between growth/biomass and fluorescence be affected by death, diauxic shift, or prolonged lag / stationary phases? See comments to the authors from Reviewer 3.

This issue is addressed in detail in response to Reviewer 3's comments. In brief, we performed interaction assays in an additional system allowing us to measure effects using both fluorescence and optical density. We saw that the effects were highly correlated, easing some of the aforementioned concerns regarding different physiological effects on the fluorescent signal. Moreover, to address concerns of autofluorescence, we ran an additional experiment in the kChip, which included wells of affecting species monocultures so we could measure their fluorescence. The difference between some affecting and focal species was smaller than anticipated with autofluorescence signals even surpassing some of the focal's signals in a handful of cases. In response to this issue, we filtered the dataset to only include affecting species with an autofluorescence signal at least fivefold lower than that of the focal species. This allows us to measure effects of more than 1.5, while still retaining a large dataset.This led to a significant reduction in the size of our dataset, and affected the different focal species unevenly (those focal species with weaker fluorescent signals were disproportionately affected). This reduced the size of our dataset by nearly half (from ~14,000 to ~8,000) as the removal of a single species exponentially affects the amount of pairs and trios that could be used. Additionally, artifacts in the data caused by autofluorescence would weaken negative effects (most of the effects found in our study), and not push observed effects towards the strongest model. Though we ultimately decided to take a conservative approach regarding which data to include, the qualitative claims of our study did not change from those made when using all the collected data.

The following was added in Appendix 1:

"Additionally, an experiment was carried out in the kChip to measure autofluorescence of affecting species. This setup was identical to the droplet preparation and culturing protocol detailed above, except that cultures were not mixed with the focal species prior to droplet generation. In this setup each droplet contains a single species, and wells contain one or two species (depending on whether the droplets were from the same or different cultures). Isolates were only used with focals whose monocultures were at least five times larger than the isolates autofluorescence signal, allowing to measure effects of at least -1.5. Full datasets without autofluorescence filtering can be seen in Appendix 1, Figure 3." (Appendix 1, Lines 37-44)

Reviewer #1 (Recommendations for the authors):My only suggestions are on writing. Certain key aspects of experiments can be outlined in the main text. For example, when pairs were introduced, did the inoculum for each strain get halved?

We apologize this was not explained clearly enough in the original manuscript. The affecting species densities are halved when introduced to the well, but as the focal species is present in both droplets, its density is not affected. Depending on the combination of types of droplet in each well, density ratios (of the affecting species to the focal) for pairs could be identical or half of the single species coculture (see Figure 1—figure supplement 1 below). We observed that different starting densities did not affect the interactions significantly, and as such, different starting ratios of the same communities were unified for downstream analysis. We have added the following sentence to main text as well in order to clarify the starting densities:

“Cultures were normalized and mixed after pre-growth, such that the starting densities in the kChip were approximately 1:1 for all species in wells containing two droplets and two affecting species, but ratios varied in three droplet wells (see Materials and methods, Figure 1—figure supplement 1).”

Furthermore, we revised the Materials and methods:

“Each well contained droplets with the same focal species such that with this setup, in a well containing two droplets of different affecting species, the starting OD_600_ of each species is 0.01 (as each affecting species is diluted by the other droplet in which it is not contained, but the focal species is not). In wells with three droplets, the starting ratio of the focal to each affecting species (assuming different species in each droplet) was 3:2. When one of the droplets contains a monoculture of the focal or is empty, or more than one droplet contains the same affecting species, these ratios change (see Figure 1—figure supplement 1).” (Lines 430-436)

Lastly, Figure 1—figure supplement 1: has been modified to better reiterate this point both graphically and the figure caption has been corrected to to match the text in the methods section.

Figure 3A graphic is unclear, and can be made much clearer by plotting mono effects and predicted coeffects under the three models.

We have made multiple changes to Figure 3, including to panel A as suggested here and by reviewer 2. Specifically we have added axes titles and a legend to panel A. This is further detailed and shown below in response to Reviewer #2's comments.

Reviewer #2 (Recommendations for the authors):Figure 3: (A) I think this panel could be made clearer. What does the green bar represent? It is not explicitly clarified in the caption. I think it depicts the growth of the green "test" bacteria? Maybe clarifying this in the caption would help. Also, why not add the equation representing the interaction?

We have added the equations for each model to the panel, a title to the y-axis, and a legend explaining each category on the x-axis. Additionally, We better clarified this panel in the figure caption (see revised figure below).

Figure 3: (B). The data points are so tiny it is difficult to see them. Maybe using larger dots (I understand there are a lot of them, though. I wonder if there is a clearer way to plot this data). Also the color choice is not the friendlies to color-challenged readers like this reviewer. I had trouble in particular distinguishing the pairs when one species had a positive while the other one had a negative effect from those that were both negative.

We agree that it is difficult to see the individual data points. Unfortunately, we were unable to come up with a better way of plotting the data – we have tried several alternatives and we feel that they reduce the amount of information we are able to convey more than they increase the clarity of the data shown. We have adopted your suggestions to (A) make the dots larger and (B) change the color map (for this and all similar plots in the paper) to improve understanding and accessibility for all readers. Below are the relevant panels from Figure 3:

Lines 167-169: The authors introduce the "mean effect model". What is the theoretical justification for including this model in the analysis? I mean, in terms of ecological theory? The additive model is justified e.g. in that it is the typical assumption in Lotka-Volterra (and also considered in the antibiotic combination literature cited in the paper). But how about the mean model? It would help if the authors explained/justified the theoretical basis for this. Otherwise it feels a bit random, they could have taken the median, or the square root of the variance, or…

The justification for this model is not immediately based in ecological theory, but rather offers an intuitive null model for the reader, as the mean is often applied when combining different functions. This is a simplistic phenomenological model that assumes effects of different species would be diluted in the presence of a third species. We have stated so explicitly in the main text to avoid confusion of future readers:

"The mean model represents a simple phenomenological model which assumes that the effects of different species will be diluted in the presence of a third species". (Lines 188-191)

Lines 221-224: The authors write: "We further explored this model by basing trios' data not on the additive, mean, or strongest values of the effects of individual species, but on those of the joint effects of the three pairs comprising each trio (the effects of single species and pairs were measured independently again in this experiment, see Materials and methods, Figure S1)." I have read this sentence five times and I am still not sure what the authors meant

We changed the above sentence to the following to improve clarity:

"We further explored the additive, mean, and strongest models in trios by basing the model on effects of the three pairs comprising each trio (i.e. joint effect of AB, AC, BC to predict effect of ABC), as opposed to only using single effect data (i.e. effect of A, B, C on their own) (Figure 4A,B). The effects of single species and pairs were measured again independently in this experiment (see Materials and methods, Figure 1—figure supplement 1)." (Lines 262-266)

Lines 296-300. The authors may want to more explicitly bring up in their discussion that the predictive ability of the "Dominance", strongest-species model, is significantly worse when they try to predict the effect of a trio than when they try to predict the effect of a pair. This would suggest some caution in the reach of their conclusions, as it is possible that the predictive power of a single species will get worse and worse as the diversity of the community increases. Which, by the way, would not be surprising I think and should not be held against their findings, but I still think it would be good to qualify their statements about the potential reach of their conclusions for the bottom-up prediction of population-level interactions in complex communities. While they do state that "further work is needed…" to figure out if they results hold in more diverse communities (Lines 294-96) I felt that the limitations of the study could be written in a sharper manner.

We have added the following sentence to the discussion, to bring attention to this concern:

"Specifically, as we saw a decrease in prediction power from pairs to trios, exploring this model with more diverse communities is of particular interest." (Lines 361-363)

Finally: I was curious if the authors have considered a model where one of the species is dominant in a pair, but the one that dominates is not necessarily the one with the strongest effect? For instance, is it possible that when A is grown with either B alone or C alone, the suppression of growth from B is stronger than the suppression of growth from C. Yet, in the presence of both B and C, the suppression of growth is exactly the same as that by C (or just closer to C than B)? Do the authors see this in any of their pairs? If so, how many?

We have considered additional models, but in the end decided to focus our predictions using only the information of single effects, to root our models in easily obtainable information. In Author response image 1 we explore two additional models- a weakest effect model and dominant effect model. The weakest effect model assumes that the species with the weaker effect will dictate the combined effect of both species. As is to be expected, due to the strongest model's accuracy, this model was not particularly accurate- and we saw that the weakest effect was only more accurate than the stronger effect model in 23% of cases. It is also worth noting that the accuracy of the weakest model dropped rapidly as the difference in effect size between species grew. This points to the option that the cases in which it was more accurate could be due to minimal differences between strongest and weakest predictions, where measurement noise could tip the balance. We found only 14 pairs where there were large differences (>1) between the single affecting species and the difference in model accuracy (for weakest and strongest). Of these, all were with EC as the focal species, and of the 14, 9 included a specific specie (i.e. *P_rhodesiae2*) as the stronger single effect, but less accurate predictor.

The dominant effect model chooses whichever single effect is closer to the combined effect to pick as the predicted effect. Though this generates a more accurate fit, this model is not predictive, as it requires the combined effect as input to choose which single species effect to use as the 'prediction'.

**Author response image 1. sa2fig1:** Exploring additional prediction models including weakest effect and dominant effect. (**A**) Correlation between four different models for how single species effects combine into pair effects and the experimental data, with their respective normalized root mean squared error. As previously described, the additive model assumes that the effects of each species will accumulate. The strongest effect model assumes that whichever species had a stronger effect on its own will determine the joint effect when paired with an additional species. By contrast The weakest effect model assumes that whichever species had a weaker effect on its own will determine the joint effect when paired with an additional species. The dominant model uses whichever single species effect is closer to the combined effect as it's prediction. (**B**) The accuracy of each model as a function of the difference between the sizes of effect of each individual species within the pair (similar to Figure 3C from main text).

Reviewer #3 (Recommendations for the authors):The metric of growth for the focal species was fluorescence. This can be a risky measurement, because other species could autofluorescence in the emission spectrum. Additionally, fluorescent proteins can continue to fluoresce after cell death and lysis (we have personally observed this after phage infection and antibiotic treatment). I think the paper could use a test to verify that fluorescence was an unbiased proxy for growth.

This point brought up by both reviewers 1 and 3 is of great importance. Experiments to test this matter, on some of the species used in this study, and in this system, have been previously performed by Kehe *et al.* (See doi:10.1126/sciadv.abi7159, specifically figure S8) and these experiments found a high correlation between fluorescent signal, OD_600_, and CFU counts. Nonetheless, we performed an interaction assay measuring fluorescence and OD_600_ simultaneously, so they could be directly compared. We achieved these simultaneous measurements by utilizing the HTD Equilibrium Dialysis System, which enabled us to measure both fluorescence and. OD_600_ for each of the two interacting species when they are cocultured in the same well. Briefly, this system is similar to a 96 well plate, but each well contains a dialysis membrane splitting the well into two halves (similar to the following protocols: https://doi.org/10.1101/2021.01.07.425753, https://doi.org/10.1371/journal.pone.0182163). Bacteria cannot pass through the membrane, but can interact chemically as dividing membrane has 1 μm pores. In each well there was a focal species and an isolate, on separate sides of the membrane, after 24 hours of coculture the OD_600_ and fluorescence was measured for each species separately. We performed this assay for 6 single species with each focal, in triplicate. The correlation between the effect when measured by fluorescence and OD_600_ was strong (r=0.71, p=1e-6). The correlation between the effect when measured in the kChip and the HTD Equilibrium Dialysis System was weaker (r=0.59, p=0.001), we believe this may be caused by the considerable differences in the physical conditions between these two setups. Overall, these results show that fluorescence is a good proxy for change in biomass in our interactions, acting similarly to OD_600_ measurements.

We have added the following paragraphs and figure to the Materials and methods:

"*Focal species were transformed with commercially available plasmid pMRE132 containing GFP2 as described by Kehe et al. 2021. Fluorescences has some caveats as a measurement for biomass, as fluorescent signal is not always directly proportional to biomass, expression levels can vary in different physiological states, and signal stability can differ between strains. Nonetheless, as described in Appendix 1, we show that effect sizes assayed using fluorescence and standard OD600 are well correlated (Appendix 1, Figure 1).* (Lines 452-457)

*"Additionally, isolates were only used with focals whose monocultures were at least five times larger than the isolates autofluorescence signal, allowing to measure effects of at least -1.5. Full datasets without autofluorescence filtering can be seen in Appendix 1, Figure 3. Importantly, affecting species autofluorescence would weaken measured negative effects, and would not systematically generate artifacts that support the strongest effect model."* (Lines 462-467)

In addition, we added the following explanations of the experiments carried out to address these issue as Appendix 1:

"Fluorescence assays

To test the accuracy of using fluorescence to assay interactions, we performed the following experiment correlating effect size as measured by fluorescent signal to effect size as measured by OD600. Bacterial strains were seeded from -80 stock directly into 0.5 ml LB medium in a 96 well plate, and grown overnight at 30°C at 900 RPM (on a Titramax 100). Cells were washed 3 times by centrifugation at 3600 rcf for three minutes, removal of supernatant, and resuspension in M9 minimal media (with the addition of 1% [w/v] glucose). All cultures were normalized to 0.02 OD600. HTD96b plates (HTDialysis, Gales Ferry, CT, USA) with membranes containing 1 μm pores splitting each well were seeded with 150 μl affecting species and focal species cultures on opposite sides of the membrane. After a 24 hour growth period at 30°C, shaking at 600 RPM, 100 μl of culture for each side of each well was transferred to a standard 96-well plate and OD600 and fluorescence were measured (Appendix 1, Figure 1). Each interaction was measured using three technical replicates.

To ensure that model accuracy was not influenced by (fluorescent) measurement limitations, we analyzed the competitive effects of models with predictions limited to the range of minimal observed measurements (as we know the maximal measurements were not near saturation). This affected only the additive model (which was the only model that could predict effects stronger than those observed), and its accuracy was improved, but it was still less accurate than the mean and strongest models (Appendix 1, Figure 2).

Additionally, an experiment was carried out in the kChip to measure autofluorescence of affecting species. This setup was identical to the droplet preparation and culturing protocol detailed above, except that cultures were not mixed with the focal species prior to droplet generation. In this setup each droplet contains a single species, and wells contain one or two species (depending on whether the droplets were from the same or different cultures). Isolates were only used with focals whose monocultures were at least five times larger than the isolates autofluorescence signal, allowing to measure effects of at least -1.5. Full datasets without autofluorescence filtering can be seen in Appendix 1, Figure 3.

I am confused by the densities that the species start at. In the methods, it says the focal and affecting species had starter cultures that were 2-fold different in concentration, yet were mixed 1:1, and ended up with a 1:1 ratio. How is this possible? Supp Figure S1 did not help me understand this.

We addressed this issue in response to Reviewer 1, who also had difficulty with this matter. In retrospect, both our wording and graphical explanations were lacking, but have been thoroughly revised to address the issue in the new version of the manuscript.

It was surprising to me that inoculation density had no effects. This makes me wonder whether the interactions observed in this study are dominated by primary metabolic competition, because density effects are very common when allelopathy occurs. If this is true, it restricts the generality of the results, and is worth being discussed. Related to this, antibiotic resistance was measured, but what about potential to secrete antibiotics?

In our discussion we address the reasons we do not believe that the interactions are dominated by metabolic competition (e.g. no hierarchy, weak correlation between effect and growth rate or carrying capacity). Unfortunately, testing for antibiotic secretion specifically is not possible in our current experimental setup. Though whole genome sequencing could potentially offer insight into this matter, we don't currently have that data for the species in our study. We have altered the following paragraph in the discussion to better address how different kinds mechanisms of competition could affect the manner in which they combine:

"As we did not measure the abundance of all species in each community (only the focal), we cannot disentangle interaction modification (changes in per capita effect of specific species), from interaction chains (affecting the amount of an affecting species, and as such its effect on the focal), and further work is needed in order to pinpoint the exact mechanism(s) leading to the dominance of the strongest model for negative effects in our system. We also note that it is possible that the manner in which effects combine is affected by the mechanism of interaction; For instance, previous studies have shown that interference competition can combine additively, or even synergistically, results not seen in our work (Tyc et al. 2014; Westhoff et al. 2021)."

(Lines 336-344)

I think the density effects should be measured with nRMSE, or even absolute difference from y=x, because there could be a strong correlation without the actual numbers being the same sign or magnitude. For example, in S11-B, most of the datapoints appear below y=x, until the effects are near zero, suggesting an effect-size-specific effect of density.

Values in the figure legend of Figure 1—figure supplement 2 have been changed from r^2^ to nRMSEs as suggested. The values are 0.22, 0.16, and 0.24 for panels A, B, and C respectively, and reflect what we maintain to be quite strong correlations for different starting densities, as well as a good absolute fit to y=x. Though there may be effect size specific behaviors (in panel B- 3:1 by 1:1 starting ratios), there is not enough data to draw significant conclusions about this at present.

More discussion could be given on what a "meaningful" difference in nRMSE is.

Though we do not discuss this explicitly in the manuscript, we agree that there is much to be said on the matter of statistically and biologically relevant differences in nRMSE. In the new version of the manuscript we focus our message around the competitive interactions, where there are large differences in the nRMSEs, and soften our claims regarding other interaction types where nRMSEs are closer to one another, to avoid exaggerated claims based on small nRMSE differences.

More details on how the resampling during the bootstrap procedure was done is warranted.

We have revised the relevant paragraph in the Materials and methods sections to include a more comprehensive explanation:

"Root mean square error measuring the accuracy of each model was normalized to the interquartile range for each dataset. Normalized root mean square error median and interquartile ranges were calculated via bootstrapping. The dataset from each focal was sampled 1000 times with replacement. Sampling was done for individual effect measurements (specific wells), and median effect sizes for species, pairs, and trios were recalculated from these sampled datasets. The sampled datasets from each focal were assembled into 'full' datasets (containing all focals) from which nRMSEs were calculated. The median and interquartile range of the normalized root mean square errors were calculated from the 1000 sampled datasets' values." (Lines 509-517)

It isn't clear to me how Figure 1C is supposed to show that different models were used-perhaps drop this and just leave this explanation for Figure 3A, which is very clear.

We agree that the panel title did not match the graphic well, despite this, we feel the panel is important for the overview of our pipeline. As such, we have changed the panel title to "*Single species data were used to predict combined effects*".

Suggestion for additional analysis using the trait data: this dataset seems perfect for using something like a random forest or other continuous-response, "open box" machine learning approach to agnostically ask whether the trait measurements can be used to predict effect when all the measurements are used, rather than summaries of the measurements in the distance metrics.

As mentioned earlier in response to the last comment of your (Reviewer 3's) public review, similar studies have been done in our research group, but we feel such analyses are beyond the scope of our current study. We do hope to apply principles learned from the other studies to our dataset in the future.